# Radiological appearance and lung function six months after invasive ventilation in ICU for COVID-19 pneumonia: An observational follow-up study

Ylva Konsberg[1], Pawel Szaro[2,3], Anders Aneman[1,4,5,6], Sanna Kjellberg[7], Nektarios Solidakis[2,3], Sara Svedlund[8], Bengt Nellgård[1], Keti Dalla[1] *

1 Department of Anesthesiology and Intensive Care, Institute of Clinical Sciences, the Sahlgrenska Academy at the University of Gothenburg, Mölndal, Sweden, 2 Department of Radiology, Institute of Clinical Sciences, Sahlgrenska Academy, University of Gothenburg, Gothenburg, Sweden, 3 Department of Radiology, Region Västra Götaland, Sahlgrenska University Hospital, Gothenburg, Sweden, 4 Intensive Care Unit, Liverpool Hospital, South Western Sydney Local Health District, Sydney, Australia, 5 South Western Clinical School, University of New South Wales, Sydney, Australia, 6 Ingham Institute for Applied Medical Science, Sydney, Australia, 7 School of Public Health and Community Medicine, Institute of Medicine, Sahlgrenska Academy University of Gothenburg, Göteborg, Sweden, 8 Department of Clinical Physiology, Sahlgrenska Univsersity Hospital, Gothenburg, Sweden

* keti.dalla@gu.se

**Data Availability Statement:** All relevant data for this study are publicly available from the Swedish

## Abstract

### Background

Respiratory functional sequelae in COVID-19 patients admitted to the intensive care unit for invasive ventilation are sparsely reported. The aim of this study was to investigate the radiological lung appearance, lung function and their association at 6 months after hospital discharge. It was hypothesized that the degree of pathological morphology on CT scans would correlate with lung function at the time of follow-up.

### Methods and findings

In this single-centre prospective observational study, 86 from 154 patients admitted to ICU due to COVID-19 between March 2020 and May 2021 were followed up at 6 months post discharge with computed tomography (CT) of the chest and pulmonary function tests (PFTs). The PFT results were expressed as z-scores calculated as the difference between the measured and predicted values divided by the standard deviation obtained from a reference population. Correlations were evaluated by Spearman's rho including the 95% confidence interval. Pathological changes on CT were found in 78/85 participants with fibrous parenchymal bands being the most prevalent finding (91%) followed by traction bronchiectasis (64%) and ground glass opacities (41%). Sixty-five participants performed PFTs, and a restrictive pattern was the most prevalent abnormality (34%). Diffusing capacity of the lung for carbon monoxide (DLCO) was reduced in 66% of participants. The CT severity score weakly correlated with forced vital capacity (FVC) z-score (0.295, p = 0.006), DLCO z-score (-0.231, p = 0.032) and alveolar volume (VA) z-score (0.253, p = 0.019).

National Data Service repository (https://doi.org/
10.5878/c4nr-bt90).

**Funding:** Prof. Bengt Nellgård ALF ("Avtal om
Läkarutbildning och Forskning") connected to the
University of Gothenburg https://www.
alfvastragotaland.se/ This sponsor played no role in
the study design, data collection and analysis,
decision to publish, or preparation of the
manuscript. Stohnes Foundation https://www.
stohnesstiftelse.se/en-GB This sponsor played no
role in the study design, data collection and
analysis, decision to publish, or preparation of the
manuscript. Gamla Tjänarinnor https://www.
stiftelseansokan.se/Pages/GamlaTjanarinnor.aspx
This sponsor played no role in the study design,
data collection and analysis, decision to publish, or
preparation of the manuscript.

**Competing interests:** The authors have declared
that no competing interests exist.

## Conclusions

Most patients showed persistent radiological abnormalities on CT and reduced lung volumes, impaired diffusion capacity and patterns of restrictive lung function at 6 months post discharge from the ICU. The correlations between abnormalities on CT and lung function tests were weak. Further, studies with a long-term follow-up of lung function in this group of patients are needed.

## Introduction

A global survey of published incidence and outcome reports early in the SARS-CoV-2 (COVID-19) pandemic found that about a third of hospitalised patients developed acute respiratory distress syndrome (ARDS) and around a quarter required admission to the intensive care unit (ICU) [1]. A significant number of COVID-19 patients admitted to ICU have residual pulmonary lesions 3 to 12 months following ICU discharge with fibrotic changes being most frequent [2–4]. Reduced diffusing-capacity of the lung (DLCO) has also been reported following recovery from COVID-19 pneumonia [5–10]. A number of studies have presented data on the respiratory effects of critical COVID-19 disease during the first year of recovery after discharge from hospital [6, 7, 9, 11–17]. However, only a few of these studies have exclusively included ICU-patients being mechanically ventilated [6, 10, 11, 17]. Understanding the long-term sequelae of COVID-19 in patients receiving mechanical ventilatory support in ICU is important in order to provide adequate information and prognostication to patients and families. While imaging is widely used to assess COVID-19 related pathology, only a minority of patients eventually undergo detailed lung function testing. The extent to which imaging modalities may be extrapolated to indicate functional outcomes remains incompletely understood.

The aim of this study was to investigate the radiological lung appearance, lung function and their association 6 months after hospital discharge, in COVID-19 patients who underwent invasive ventilation in the ICU. It was hypothesized that the degree of pathological lung morphology on CT scans would correlate with impairment of lung function at the time of follow-up.

## Methods

### Study design and study population

This single-centre, prospective observational study was conducted at the Department of Anaesthesiology and Intensive Care at Sahlgrenska University Hospital/Mölndal. Ethical approval for the study was granted (National Ethical Authority, approval number Dr 2020–03660). The clinical trial number was NCT 05299346. Patients were included if ≥18 years with confirmed COVID-19 pneumonia by a positive SARS-CoV-2 PCR nasopharyngeal test, and admitted to ICU for invasive respiratory treatment between the 24th of March 2020 and 20th of May 2021. Patients were excluded if not surviving to hospital discharge, declined participation, or had moved outside Swedish healthcare.

Eligible patients were contacted by telephone and/or mail and were offered a follow-up examination at around six months after ICU discharge. All patients gave written informed consent prior to follow-up and all study procedures were free of charge. The follow-up visits occurred amid ongoing pandemic social restrictions.

## Clinical data collection

Patient characteristics included age, gender, body mass index (BMI) and medical comorbidities (hypertension, cardiovascular disease, diabetes, chronic obstructive pulmonary disease (COPD), asthma, smoking history, drug history). Treatment data included length of stay in ICU (LOS-ICU), the number of days with invasive mechanical ventilation, and the use of prone positioning. Laboratory data included arterial blood gases collected from patients while being in the ICU. Patient, treatment and laboratory data were collected from the patients' medical records retrospectively. At the follow-up visit, each patient underwent a 256-slice multidetector CT of the chest without contrast and PFTs including dynamic spirometry and DLCO. A subgroup of patients performed body plethysmography at time of follow-up. Patients also performed a Time Up to Go (TUG) test, which is a test investigating physical fuction. The TUG-test measures the time a person takes to rise from a chair, walk three metres, turn around 180 degrees, walk back to the chair and sitting down again while turning 180 degrees [18].

## Radiological data

All CT images were examined by two radiologists at Sahlgrenska University Hospital/Mölndal. Regarding the review process for the examinations, they were examined collaboratively with careful consideration by both parties. The review followed established protocols to ensure that the interpretation of the results was accurate and thorough. The following radiological variables were included as parenchymal lesions [23–25]: ground-glass opacity (opacification or increased attenuation due to air displacement and inflammation, caused by fluid, airway collapse or fibrosis) [19]; fibrous parenchymal bands (bands, running through the lung parenchyma and usually extending from a visceral pleural surface, suggesting pleuro-parenchymal fibrosis, and frequently related to distortion of the lung parenchyma); traction bronchiectasis (irreparable distension of bronchi and bronchioles in areas of pulmonary fibrosis or distorted lung parenchyma architecture); and air space opacification/pulmonary consolidation (filling of the tracheobronchial tree with material that attenuates more than the adjacent lung parenchyma, causing obscuration of pulmonary vessels). The parenchymal lesion distribution was assessed as being peripheral, central, or both. The grade of lobe involvement was assessed using a visual scale: grade 1 (1–5%), grade 2 (6–25%), grade 3 (26–50%), grade 4 (51–75%) and grade 5 (more than 76%).

Finally, a CT severity score, was calculated proportional to the degree of lobe involvement with points awarded according to: grade 1 = 1 point, grade 2 = 5 points, grade 3 = 10 points, grade 4 = 15 points, and grade 5 = 20 points, generating a semiquantitative scale adapted from Pan et al. [3, 20].

## Spirometry, diffusing-capacity for carbon monoxide and plethysmography data

Dynamic spirometry and DLCO was performed using a Jaeger MasterScreen® PFT system (Care Fusion, Würzburg, Germany), in accordance with American Thoracic Society/European Respiratory Society (ATS/ERS) standards [21, 22]. Spirometry derived forced expiratory volume in one second ($FEV_1$), forced vital capacity (FVC) and their ratio ($FEV_1$/FVC) were measured in all 65 patients. Slow vital capacity (SVC) was measured during the follow-up period between the 27th of October 2020 and 3rd of December 2020 in 30 study patients and volume capacity (VC) in 33 patients. Plethysmography was made available as part of follow-up between the 14th of September 2021 and 14th of October 2021, for the remaining 29 study

patients in whom total lung capacity (TLC), functional residual capacity (FRC) and residual volume (RV) were measured. Diffusing-capacity of the lung and the alveolar ventilation were possible to be assessed, in 62 patients, using a single-breath DLCO technique, where DLCO represents the amount of CO that is transferred from the alveoli to the lung capillaries, expressed as mmol×min-1×kPa-1. Furthermore, the DLCO in relation to alveolar lung volume (VA), was reported as DLCO/VA. All DLCO values were corrected for hemoglobin concentration. Each spirometry and DLCO examinations were assessed by a specialist in respiratory physiology.

Lung function results were expressed as z-scores and calculated as the difference between the measured value and predicted value divided by the standard deviation obtained from a reference population [23–25]. The z-score is also known as "standard score" that is a measure of how many standard deviations below or above the population mean a raw score is, i.e. a z-score < -1.96 places the raw score in the lowest 2.5th percentile of the refence population.

## Statistical analysis

For this observational study, it was determined that a total sample size of 78 patients resulted in a power of 90% to detect a medium effect size (Cohen's d) of 0.65 corresponding to a correlation factor of 0.31 between CT severity scores and lung function variables in a two-sided t-test with a type I error of 5% [26].

Categorical variables were described by numbers and percentages. Continuous variables were described by mean ± standard deviation (SD) or as median and interquartile range (IQR) for normally and non-normally distributed data. The correlations between pulmonary function test results, CT severity score, duration of mechanical ventilation, mean P/F ratio during the first week in the ICU and patients' functional capacity measured with the TUG test, were evaluated using Spearman's rho. The primary outcome was the correlation between the CT severity score and z-scores of lung function measurements. Secondary outcomes were the correlations between lung function and duration of mechanical ventilation, P/F ratio and TUG results.

Spirometry, plethysmography and DLCO tests could not be performed in all patients (24–36% missing data). This degree of missing data is not unexpected given the pragmatic nature of the study and reflects pandemic social restrictions precluding follow-up visits as well as the constraints on the use of potential aerosol generating procedures and healthcare resources. Missing data were assumed to be missing at random (MAR) and replaced by imputed data using the MICE package in R (version 3.14.7). Multiple imputations by chained equations were performed by fully conditional specification for multivariate missing data using 20 iterations [27]. All variables in the complete model were used but derived variables were created after imputation. The predictive mean model was used for numerical data. The quality of the pooled imputations was assessed by visual inspection to confirm overlap on the strip and density plots. Results are reported based on complete case data and imputed data separately that are compared in a sensitivity analysis by unpaired t-tests and reported by the mean difference and its 95% confidence interval (CI).

Statistical analyses were performed using SPSS Statistics version 28.0.1.0 (IBM Corporation, Armonk, New York, USA) and the R statistical software version 4.0.3 (R Foundation for Statistical Computing, Vienna, Austria). All significance tests were two-sided with statistical significance set at p-value <0.05.

## Results

### Patient characteristics

A total of 154 patients received invasive ventilatory treatment in the ICU for COVID-19 pneumonia during the study period and 86 of these attended the follow-up visit (Fig 1). CT

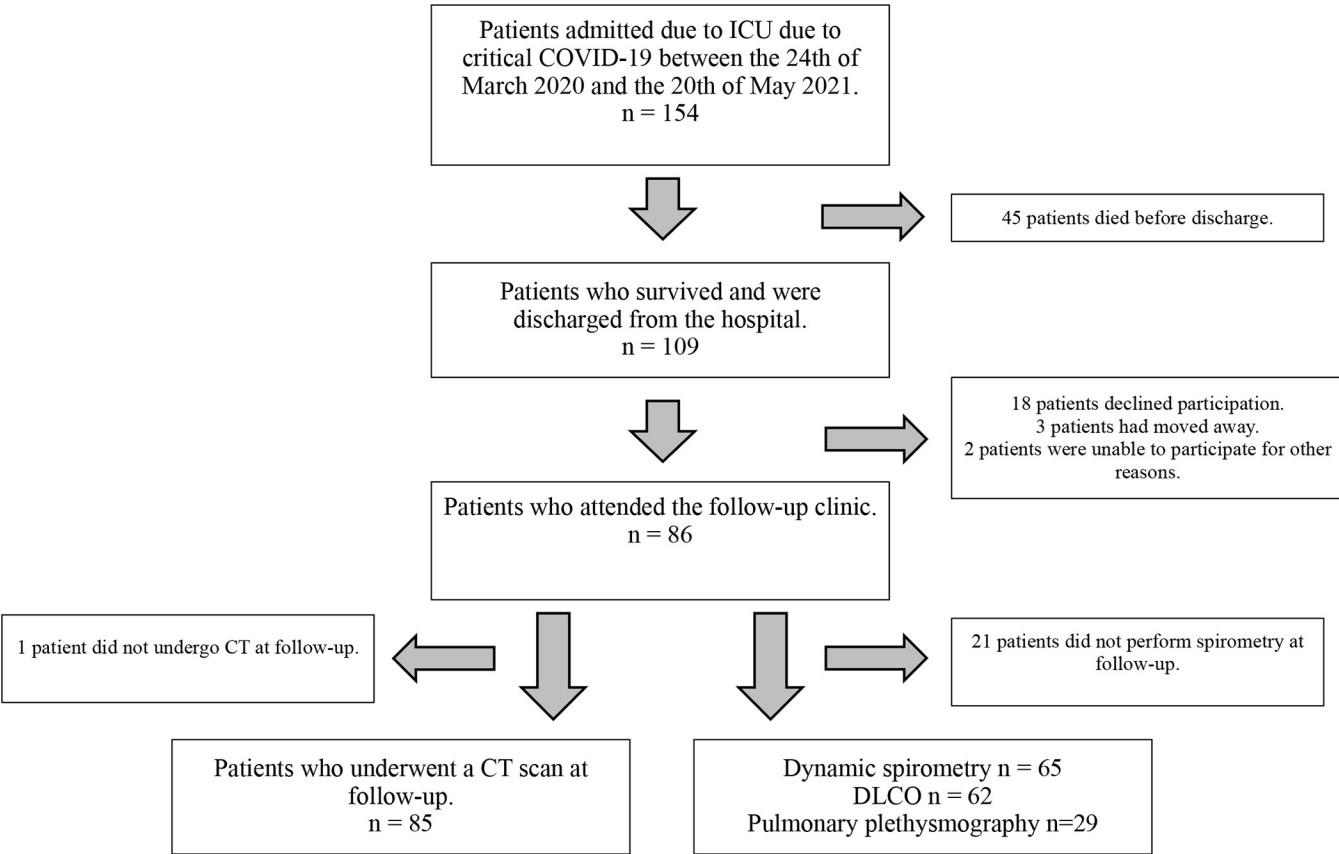

**Fig 1. Flow-chart showing patient selection and participation in the follow-up clinic.** 86 of the 154 patients who were treated in the ICU due to critical COVID-19 during the inclusion period attended the follow-up clinic.

scanning was performed in 85 (99%), dynamic spirometry ($FEV_1$, FVC and $FEV_1$/FVC) in 65 (76%) and DLCO in 62 (72%) of patients. During the two phases of the study, dynamic spirometry (SVC) was performed in 30 (75%) patients during early follow-up and plethysmography (TLC, FRC and RV) in 29 (63%) patients during late follow-up. Vital capacity (VC) was performed as part of plethysmography during the late follow-up in 33 (72%) patients. The mean time to follow-up after discharge from the ICU was 197±43 days.

Patient characteristics are reported in Table 1. The mean age was 62 years with a majority being males 61 (71%). The mean BMI at ICU admission was 31 kg/m$^2$ and the most common comorbidity was hypertension (42%). The mean P/F ratio during the first week in ICU was 175 ±44 mmHg; (n = 78). Most of the patients (77%) had a mean P/F ratio between 100 to 200 mmHg, whereas no one had a mean P/F ratio below 100 mmHg.

## Radiology

Radiological lesions were found in 78 (92%) patients, with fibrous parenchymal bands (91%), traction bronchiectasis (64%) and ground glass changes (41%) far more common than other pathological changes (Table 2).

There were minor differences in the distribution of lesions between the five lung lobes, while pathological changes were most frequently noted (73%) in the left superior lobe of the lung. The median CT severity score per patient for the whole lung was 5 (IQR 2 to 5), with

**Table 1. Patient characteristics, clinical and laboratory data in ICU and at time of follow-up (n = 86).**

| Patient characteristics | |
|---|---|
| **Age** | |
| Mean | 62 (28) |
| **Gender** | |
| Male, n (%) | 61 (71) |
| **BMI on admission to the ICU, n = 85 (kg/m2)** | |
| Mean | 31 (6) |
| **BMI at time of follow-up dynamic spirometry, n = 65 (kg/m2)** | |
| Mean | 31 (5) |
| **LOS-ICU (days)** | |
| Mean | 22 (15) |
| **Days of mechanical ventilation** | |
| Mean | 19 (15) |
| **PaO2/FiO2 ratio during first 7 ICU-days, n = 78** | |
| Mean | 175 (44) |
| **Prone position** | |
| Prone position ≥1 episodes during ICU stay, n (%) | 57 (66) |
| **Days from ICU-discharge to follow-up** | |
| Mean | 197 (43) |
| **Comorbidities** | |
| History of smoking, n (%) | 24 (29) |
| Asthma, n (%) | 3 (4) |
| Chronic Obstructive Pulmonary Disease (COPD), n (%) | 3 (3) |
| Diabetes, n (%) | 13 (15) |
| Hypertension (%) | 36 (42) |
| Treatment with ARB/ACEi, n (%) | 28 (33) |
| Cardiovascular disease, n (%) | 20 (23) |

For categorical variables n (%) is presented. For continuous variables, mean (SD). ICU = Intensive care unit; BMI = Body mass index; LOS-ICU = Length of stay in intensive care unit; PaO2/FiO2 = arterial oxygen partial pressure to fractional inspired oxygen; COPD = Chronic obstructive pulmonary disease; ARB = Angiotensin receptor blocker; ACEi = Angiotensin converting enzyme inhibitor

pulmonary parenchyma points equally distributed between the different lobes. The CT severity score was overwhelmingly generated by the presence of fibrous bands.

## Spirometry and DLCO

Complete data and data including imputed cases for all spirometry and DLCO measurements are presented in Table 3, alongside the results of a sensitivity analysis generated by the unpaired t-test. No significant mean difference was found between complete data and data including imputed data. Hence, results including imputed data will be discussed in this section.

The number of patients with findings lower than normal range (z-score <-1.96) are presented in Table 3. The mean $FEV_1$ z-score was -0.462; mean FVC z-score was -0.643 and the mean $FEV_1$/FVC z-score was within the normal range with only 1/86 (1%) patients having $FEV_1$/FVC z-score <-1.96. SVC z-score was < -1.96 in 9/40 (23%) patients. The mean TLC z-score was -1.619, and 20/46 (43%) patients had a TLC z-score < - 1.96. DLCO and VA z-scores

**Table 2. Type of pathological changes and anatomical distribution seen on CT chest at the time of follow-up, 6 months post discharge from the ICU (n = 85).**

| Lung pathology | Number of patients, n (%), with radiological changes |
|---|---|
| Fibrous bands | 77 (91) |
| Traction bronchiectasis | 54 (64) |
| Ground glass opacities | 35 (41) |
| Infiltrates | 3 (4) |
| Contour infiltrates | 3 (4) |
| Multiple focal areas with infiltrates | 4 (5) |
| Reverse halo sign | 2 (2) |
| Widened vessels | 2 (2) |
| Septal widening | 8 (9) |
| Lymphadenopathy | 3 (4) |
| Widened truncus pulmonalis | 6 (7) |
| Pericardial effusion | 7 (8) |
| Pleural effusion | 3 (4) |
| **Anatomical involvement** | |
| Left superior | 62 (73) |
| Left inferior | 57 (67) |
| Right superior | 60 (71) |
| Right medial | 51 (60) |
| Right inferior | 58 (62) |
| Whole lung | 79 (93) |
| Absence of radiological changes | 7 (8) |

For categorical variables n (%) is presented. Whole lung; all five lobes.

were < -1.96 in 36/86 (42%) and in 31/86 (36%) patients, respectively, while only 9/86 (10%) patients had DLCO/VA z-score < -1.96.

Clinical assessment of complete cases of pulmonary function tests and DLCO by a specialist in respiratory physiology showed that 39/65 patients had a normal pattern. A restrictive pattern was the main pathology in 22/65 patients while only two patients had an obstructive pattern. One patient had a mixed restrivtive-obstructive pattern and one patient had results that could not be clinically assessed (Fig 2A). Twenty-eight out of 62 patients had a mild reduction DLCO, 12/62 had a moderate to severe reduction while 22/62 had normal range values (Fig 2B). Thus, <50% of patients had a normal DLCO. The mildly reduced DLCO is considered when the DLCO-value are < 60% of reference value, moderate reduced 40–60% and severely reduced < 40% of reference value. The FVC was mildly reduced if < 75% of reference value, moderately reduced at 50–75% and severely reduced <50% of reference value. These definitions are based on local clinical experience with purpose of providing guidance to clinicians.

## Radiology and lung function outcomes

All results presented in this section are obtained from calculations using data including imputed cases. We found weak correlations between CT severity score and FVC z-score, $FEV_1$/FVC z-score, DLCO z-score and VA z-score (Table 4).

There were weak correlations between duration of mechanical ventilation and FVC z-score, $FEV_1$/FVC z-score and DLCO z-score and VA z-score (Table 4). No significant correlation was found between pulmonary function tests outcomes and mean P/F ratio during the first 7 days in the ICU or TUG post-COVID (Table 4).

**Table 3. Spirometry and pulmonary pletysmography results for complete and imputed data, including sensitivity analysis comparing the two datasets.**

| Measurement | Complete cases, n (%) * | Complete data *Mean (SD)* | Including imputed cases, n (%) ** | Imputed data *Mean (SD)* | Mean difference and 95% CI complete vs. imputated cases *** | P-value complete vs. imputated cases *** |
|---|---|---|---|---|---|---|
| FEV$_1$ z-score | 65 (76%) | -0.447 (1.335) | 86 (100%) | -0.462 (1.298) | 0.015 (-0.412; 0.442) | 0.945 |
| FEV$_1$ z-score < -1.96, n (%) | 65 (76%) | 10 (15%) | 86 (100%) | 12 (14%) | | |
| FVC z-score | 65 (76%) | -0.585 (1.392) | 86 (100%) | -0.643 (1.337) | 0.058 (-0.384; 0.500) | 0.796 |
| FVC z-score < -1.96, n (%) | 65 (76%) | 9 (14%) | 86 (100%) | 12 (14%) | | |
| FEV$_1$/FVC z-score | 65 (76%) | 0.173 (0.908) | 86 (100%) | 0.230 (0.921) | -0.058 (-0.355; 0.239) | 0.702 |
| FEV$_1$/FVC z-score < -1.96, n (%) | 65 (76%) | 1 (2%) | 86 (100%) | 1 (1%) | | |
| SVC z-score | 30 (75%) | -0.914 (1.753) | 40 (100%) | -0.681 (1.711) | -0.234 (-1.067; 0.600) | 0.578 |
| SVC z-score < -1.96 n (%) | 30 (75%) | 7 (23%) | 40 (100%) | 9 (23%) | | |
| DLCO z-score | 62 (72%) | -1.779 (1.517) | 86 (100%) | -1.686 (1.409) | -0.058 (-0.625; 0.509) | 0.869 |
| DLCO z-score < -1.96, n (%) | 62 (72%) | 27 (44%) | 86 (100%) | 36 (42%) | | |
| VA z-score | 62 (72%) | -1.611 (1.621) | 86 (100%) | -1.480 (1.500) | 0.136 (-0,422; 0.693) | 0.955 |
| VA z-score < -1.96, n (%) | 62 (72%) | 24 (39%) | 86 (100%) | 31 (36%) | | |
| DLCO/VA z-score | 62 (72%) | -0.575 (1.212) | 86 (100%) | -0.595 (1.142) | 0.008 (-0.590; 0.605) | 0.908 |
| DLCO/VA z-score < -1.96, n (%) | 62 (72%) | 7 (11%) | 86 (100%) | 9 (10%) | | |
| TLC z-score | 29 (63%) | -1.449 (1.222) | 46 (100%) | -1.619 (1.336) | 0.170 (-0.441; 0.781) | 0.581 |
| TLC z-score < -1.96, n (%) | 29 (63%) | 9 (31%) | 46 (100%) | 20 (43%) | | |
| FRC z-score | 29 (63%) | -1.384 (0.844) | 46 (100%) | -1.418 (1.026) | 0.034 (-0.420; 0.488) | 0.881 |
| FRC z-score < -1.96, n (%) | 29 (63%) | 7 (24%) | 46 (100%) | 16 (35%) | | |
| RV z-score | 29 (63%) | -0.830 (0.814) | 46 (100%) | -0.892 (1.023) | 0.062 (-0.386; 0.510) | 0.783 |
| RV z-score < -1.96, n (%) | 29 (63%) | 1 (3%) | 46 (100%) | 3 (7%) | | |
| RV/TLC z-score | 29 (63%) | -0.131 (0.926) | 46 (100%) | -0.189 (1.019) | 0.058 (-0.407; 0.523) | 0.805 |
| RV/TLC z-score < -1.96, n (%) | 29 (63%) | 0 (0%) | 46 (100%) | 0 (0%) | | |
| VC z-score | 33 (72%) | -1.156 (1.380) | 46 (100%) | -1.255 (1.598) | 0.098 (-0.588; 0.785) | 0.776 |

(*Continued*)

**Table 3.** (Continued)

| Measurement | Complete cases, n (%) * | Complete data Mean (SD) | Including imputed cases, n (%) ** | Imputed data Mean (SD) | Mean difference and 95% CI complete vs. imputated cases *** | P-value complete vs. imputated cases *** |
|---|---|---|---|---|---|---|
| VC z-score < -1.96, n (%) | 33 (72%) | 9 (30%) | 46 (100%) | 16 (35%) | | |

For categorical variables n (%) is presented. For continuous variables, mean (SD). $FEV_1$ = Forced expiratory reserve volume during the $1^{st}$ second of expiration;

FVC = Forced vital capacity; SVC = Slow vital capacity; DLCO = Diffusion capacity for carbon monoxide; VA = Alveolar volume; VC = Vital capacity; TLC = Total lung capacity; FRC = Functional residual capacity; RV = Residual volume.

\* = number of cases with complete data out of patients eligible for measurement

\*\* = number of cases with imputed data added to all patients eligible for measurement

\*\*\* = mean difference and unpaired t-test compare the complete cases vs. the complete+imputed cases

# Discussion

This study showed that at 6 months post-discharge from the ICU due to COVID-19, fibrous bands were the most commonly seen radiological changes, followed by traction bronchiectasis

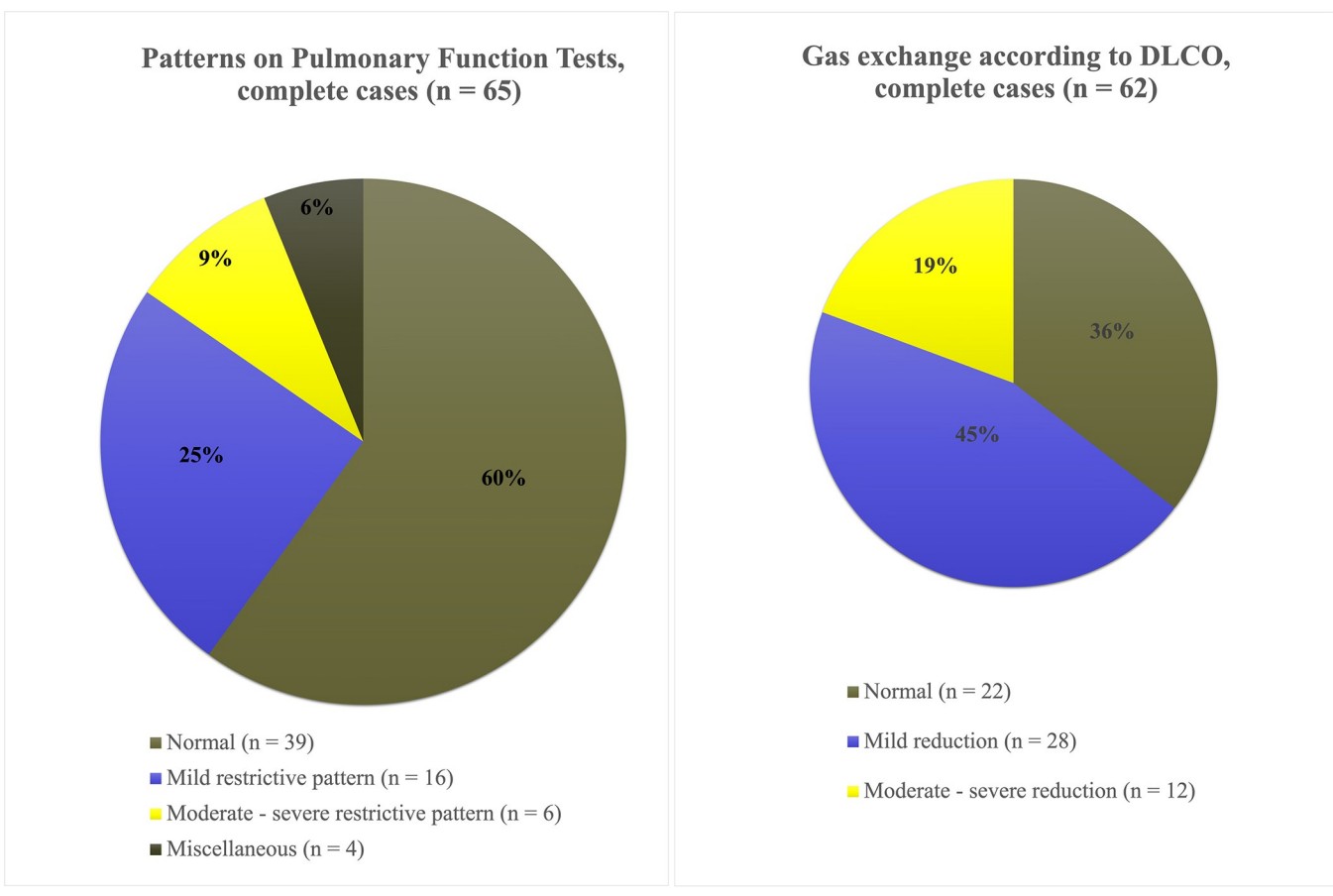

**Fig 2.** a and b. Patterns on PFTs and DLCO according to clinical assessment by a specialist in respiratory physiology. Restrictive impairment was the most frequently noted pathological pattern on PFTs and 65% of the patients had reduced DLCO at the time of follow up. Of those with moderately to severely reduced DLCO, 47% had a DLCO/VA z-score < -1.96, while only 7% of those with a mild DLCO reduction had a DLCO/VA z-score < -1.96. The terms mild, moderate and severe reductions are based on clinical assessment.

**Table 4. Spearman's correlation between pulmonary function tests and CT severity score; Duration of mechanical ventilation; Mean P/F ratio during first 7 days; and "Time Up to Go" (TUG).** The table includes correlations between both complete and imputed datasets.

| | CT severity score *Complete data* | CT severity score *Imputated data* | Duration of Mechanical ventilation (days) *Complete data* | Duration of Mechanical ventilation (days) *Imputated data* | Mean P/F ratio during first 7 ICU-days *Complete cases* | Mean P/F ratio during first 7 ICU-days *Imputated data* | TUG post-COVID (s) *Complete data* | TUG post-COVID (s) *Imputated data* |
|---|---|---|---|---|---|---|---|---|
| FEV$_1$ z-score | - 0.284* (0.022) | -0.167 (0.123) | -0.235 (0.059) | -0.151 (0.166) | -0.081 (0.541) | 0.019 (0.863) | -0.196 (0.125) | -0.099 (0.365) |
| | n = 65 | n = 86 | n = 65 | n = 86 | n = 60 | n = 86 | n = 63 | n = 85 |
| FVC z-score | -0.374** (0.002) | -0.295** (0.006) | -0.353** (0.004) | -0.233* (0.031) | -0.067 (0.611) | -0.009 (0.935) | -0.295* (0.019) | -0.166 (0.130) |
| | n = 65 | n = 86 | n = 65 | n = 86 | n = 60 | n = 86 | n = 63 | n = 85 |
| FEV$_1$/FVC z-score | 0.380** (0.002) | 0.319** (0.003) | 0.409** (0.001) | 0.288** (0.007) | -0.077 (0.558) | -0.011 (0.923) | 0.251* (0.048) | 0.141 (0.200) |
| | n = 65 | n = 86 | n = 65 | n = 86 | n = 60 | n = 86 | n = 63 | n = 85 |
| SVC z-score | -0.556** (0.001) | -0.298 0.062 | -0.493 (0.006) | -0.208 (0.197) | -0.058 (0.762) | -0.085 (0.604) | -0.293 (0.116) | -0.136 (0.404) |
| | n = 30 | n = 40 | n = 30 | n = 40 | n = 30 | n = 40 | n = 30 | n = 40 |
| DLCO z-score | -0.440** (<0.001) | -0.231* (0.032) | -0.367** (0.003) | -0.329** (0.002) | 0.075 (0.577) | 0.127 (0.242) | -0.469** (<0.001) | -0.203 (0.062) |
| | n = 62 | n = 86 | n = 62 | n = 86 | n = 58 | n = 86 | n = 60 | n = 85 |
| DLCO/VA z-score | 0.004 (0.975) | 0.008 0.941 | 0.042 (0.747) | -0.039 (0.720) | 0.027 (0.838) | 0.044 (0.688) | -0.213 (0.103) | -0.152 (0.165) |
| | n = 62 | n = 86 | n = 62 | n = 86 | n = 58 | n = 86 | n = 60 | n = 85 |
| VA (z-score) | -0.457** (<0.001) | -0.253* (0.019) | -0.408** (0.001) | -0.329** (0.002) | 0.043 (0.749) | 0.093 (0.397) | -0.363** (0.004) | -0.087 (0.429) |
| | n = 62 | n = 86 | n = 62 | n = 86 | n = 58 | n = 86 | n = 61 | n = 85 |
| TLC (z-score) | -0.260 (0.173) | -0.264 (0.077) | -0.320 (0.091) | -0.095 (0.528) | 0.061 (0.766) | -0.103 (0.495) | -0.484** (0.008) | -0.223 (0.140) |
| | n = 29 | n = 46 | n = 29 | n = 46 | n = 26 | n = 46 | n = 29 | n = 45 |
| FRC z-score | -0.037 (0.851) | -0.110 (0.468) | -0.127 (0.512) | -0.154 (0.307) | 0.073 (0.724) | 0.089 (0.557) | -0.171 (0.376) | -0.183 (0.230) |
| | n = 29 | n = 46 | n = 29 | n = 46 | n = 26 | n = 46 | n = 29 | n = 45 |
| RV z-score | -0.037 (0.851) | -0.122 (0.421) | -0.089 (0.646) | -0.161 (0.285) | 0.235 (0.248) | 0.152 (0.313) | -0.272 (0.154) | -0.151 0.321 |
| | n = 29 | n = 46 | n = 29 | n = 46 | n = 26 | n = 46 | n = 29 | n = 45 |
| VC z-score | -0.0173 (0.336) | -0.171 (0.256) | -0.331 (0.060) | -0.157 (0.298) | -0.042 (0.827) | -0.013 (0.933) | -0.343 (0.059) | -0.246 (0.103) |
| | n = 33 | n = 46 | n = 33 | n = 46 | n = 29 | n = 46 | n = 31 | n = 45 |
| RV/TLC z-score | 0.206 (0.283) | 0.248 (0.097) | 0.161 (0.507) | 0.108 (0.475) | 0.212 (0.299) | 0.189 (0.208) | 0.118 (0.541) | -0.028 (0.855) |
| | n = 29 | n = 46 | n = 29 | n = 46 | n = 26 | n = 46 | n = 29 | n = 45 |

Data are presented as r (p-value).

* indicates significant correlation. FEV$_1$ = Forced expiratory reserve volume during the 1$^{st}$ second of expiration; FVC = Forced vital capacity; SVC = Slow vital capacity; DLCO = Diffusion capacity for carbon monoxide; VA = Alveolar volume; VC = Vital capacity; TLC = Total lung capacity; FRC = Functional residual capacity; RV = Residual volume

and ground glass opacities. Pulmonary function tests showed an overall pattern of reduced lung volumes and signs of restrictive lung disease. The study also showed a decrease in DLCO, possibly linked to a reduction in lung volumes.

We found that almost all patients had radiological changes affecting all five lung fields, in line with other CT follow-up studies of COVID-19 survivors at three months post-ICU

discharge [6, 10, 28]. Generally, patients who develop CT changes post-COVID are more severely affected by the disease, with development of ARDS, and hence are more likely to require mechanical ventilation and high concentration of supplemental oxygen while hospitalized. Many authors have labeled these persistent CT abnormalities as post-COVID lung fibrosis [2, 7, 29–31]. Nabahati *et al.* observed that severe COVID-19 with consolidations and higher CT severity score on initial CT during the hospitalization was associated with increased risk of post-COVID lung fibrosis at 3 months post discharge. In contrast, Liu D *et al.* reported resolution of CT changes in 79% of the post-COVID cases at the 4$^{th}$ week of follow-up [29]. Based on our findings, in a cohort of mechanically ventilated patients, it is difficult to state if the radiological abnormalities are signs of ongoing inflammation, fibrosis, a manifestation of ventilator induced lung injury (VILI) or a combination of these conditions.

Pulmonary function tests showed that DLCO was the most frequently impaired parameter, alongside reduced VA. However, DLCO/VA was normal in most patients, theoretically indicating normal diffusing capacity and membrane function in the alveolar units participating in gas exchange. Reduction in FEV$_1$ and FVC z-scores < -1.96 were present in a relatively small proportion of our patient cohort, while TLC z-scores were more frequently reduced in the plethysmography subgroup. Upon assessment by a specialist in respiratory physiology of patients' PFTs, a restrictive pattern was the abnormality most frequently found (25% mildly restrictive; 9% moderately to severely restrictive). However, 60% of patients had a normal pattern on PFTs, which is an encouraging result considering the previous severity of COVID-19 disease in this cohort. Other studies on COVID-19 survivors, requiring ICU admission, have reported similar results regarding pulmonary function at 6 months post-discharge, and reduced DLCO has been the most frequently found impairment [5–10]. Reduced lung volumes, including reduced FEV$_1$, FVC and TLC, were found in a similar proportion of patients in these studies. Notably, the mostly normal DLCO/VA values on PFTs in this cohort highlight how DLCO depends on different physiological factors, including lung volumes [21]. This brings into question the mechanisms behind impaired DLCO in COVID-19 survivors. It has been speculated that it could be caused in part by interstitial and pulmonary vascular abnormalities [9]. However, we found that the diffusing capacity of the alveolar units participating in gas exchange was normal, indicating an impact of reduced VA and hence lung volumes on impaired DLCO.

DLCO and VA z-scores were weakly correlated to CT severity score. Furthermore, there was a weak correlation between FVC z-scores and CT severity score, possibly reflecting a relationship between the high prevalence of fibrotic changes on CT (indicating structural abnormalities in the lungs), reduced lung volumes and restrictive impairment seen in some patients. The physiological and anatomical effect, however, of COVID-19 on the lungs of survivers is still uncleared and under studied. More observational studies in this field are warranted.

In line with results of other studies [6, 7], duration of mechanical ventilation was negatively correlated to FVC, DLCO and VA z-scores. As has been pointed out by others [30, 32, 33], it is unclear to which extent this reflects a potential damaging impact of mechanical ventilation on lung function (VILI) and to what degree it is directly associated with increasing severity COVID-19. Furthermore, neuromuscular weakness post-critical COVID-19, as in non-COVID-19 ARDS, may have an impact on long term pulmonary function, although the relative effects of this in comparison to other factors has not been established [34]. In our study, we found no significant correlations between TUG, which could be viewed as a surrogate measure of patients' physical performance and status, and pulmonary function tests or DLCO using data including imputated cases.

## Strengths and limitations

A strength with our study was that we exclusively studied ICU-patients who had been mechanically ventilated during their hospital stay. The follow-up visit included assessment of anatomical changes with CT chest and lung function with PFTs and DLCO measurements. Furthermore, a specialist in respiratory physiology reviewed the results and was able to categorize patterns on PFTs, rather than just reporting the measurements in absolute values.

One important limitation of this study, however, is the lack of detailed data on specific ventilator settings for each patient, including tidal volume/kg body weight, driving pressures and plateau pressures. Furthermore, we have no data on patients' baseline pulmonary function prior to their admission to ICU due to critical COVID-19. Neither do we have "baseline" CT chest scans for the patients in this cohort, performed and assessed according to the study protocol, to which we can compare the CT chest scans performed at time of follow-up. Finally, we didn't include treatment data as Betamethasone and -or Remdesivir in statistical analysis. It was not in the aims of this study to find out the impact of these therapies on lung function at follow-up, which theoretically could affect the radiological appearance and PFTs of these patients.

## Conclusion

Mechanically ventilated ICU-patients surviving COVID-19 showed persistent radiological abnormalities on chest CT and reduced lung volumes, impaired DLCO and patterns indicative of restrictive impairment on PFTs at 6 months after discharge from ICU. Weak correlations were found between CT severity score, DLCO z-scores, VA z-scores and SVC z-score. What remains to be elucidated is whether they continue to improuve over time. Thus, further studies with a long-term follow-up of lung function in this group of patients are needed.

## Supporting information

**S1 File.**
(DOCX)

## Author Contributions

**Conceptualization:** Ylva Konsberg, Pawel Szaro, Anders Aneman, Sara Svedlund, Bengt Nellgård, Keti Dalla.

**Data curation:** Ylva Konsberg, Sanna Kjellberg, Nektarios Solidakis, Keti Dalla.

**Formal analysis:** Ylva Konsberg, Anders Aneman, Sara Svedlund, Keti Dalla.

**Funding acquisition:** Bengt Nellgård, Keti Dalla.

**Investigation:** Sanna Kjellberg, Nektarios Solidakis.

**Methodology:** Nektarios Solidakis.

**Project administration:** Keti Dalla.

**Resources:** Bengt Nellgård.

**Supervision:** Keti Dalla.

**Visualization:** Ylva Konsberg, Keti Dalla.

**Writing – original draft:** Ylva Konsberg, Keti Dalla.

**Writing – review & editing:** Ylva Konsberg, Anders Aneman, Sara Svedlund, Bengt Nellgård, Keti Dalla.

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
