## [Decision Letter · Decision Letter 0]

14 Mar 2023

PONE-D-22-31787Dear Editor,

The full title for our article is:

Radiological appearance and lung function six months after invasive ventilation in ICU for COVID-19 pneumonia: an observational follow-up studyPLOS ONE

Dear Dr. Dalla,

Thank you for submitting your manuscript to PLOS ONE. After careful consideration, we feel that it has merit but does not fully meet PLOS ONE’s publication criteria as it currently stands. Therefore, we invite you to submit a revised version of the manuscript that addresses the points raised during the review process.

We look forward to receiving your revised manuscript.

Kind regards,

Tai-Heng Chen, M.D.

Academic Editor

PLOS ONE

Journal Requirements:

Reviewers' comments:

Reviewer's Responses to Questions

**Comments to the Author**

1. Is the manuscript technically sound, and do the data support the conclusions?

Reviewer #1: Yes

Reviewer #2: Yes

Reviewer #3: Yes

Reviewer #4: Yes

Reviewer #5: Yes

2. Has the statistical analysis been performed appropriately and rigorously? 

Reviewer #1: Yes

Reviewer #2: Yes

Reviewer #3: Yes

Reviewer #4: Yes

Reviewer #5: No

3. Have the authors made all data underlying the findings in their manuscript fully available?

Reviewer #1: Yes

Reviewer #2: No

Reviewer #3: No

Reviewer #4: Yes

Reviewer #5: No

4. Is the manuscript presented in an intelligible fashion and written in standard English?

Reviewer #1: Yes

Reviewer #2: Yes

Reviewer #3: Yes

Reviewer #4: Yes

Reviewer #5: Yes

5. Review Comments to the Author

Reviewer #1: This is a single centre prospective observational study assessing lung function ( n=65 n=62 DLCO) and CT findings (n=85) at 6 months of 86 patients intubated and ventilated for covid 19. They also looked at the correlation between CT findings and lung function.

This is original research not published elsewhere.

The study has appropriate ethical approval.

Introduction describes the rationale, aims and hypothesis and is clearly and succinctly written

Methods

Clearly described with statistics

Comment 1: Were the radiologists blinded to clinical and lung function data and did they review the images independently. Please state this in methods

Missing data was imputed using MICE package

Comment 2: Table 1/2. Please put explanations at bottom of the table in the legend.

Spirometry and DLCO

Comment 3: Whilst z scores are statistically correct ways of presenting lung function data – to the majority of readers it will not be wholly understandable. Please consider presenting absolute values and % predicted values as well. Maybe as supplementary data

Comment 4: How is mild moderate and severe reductions in DLCO or FVC defined

Figures 2.1 and 2.2 How are mild moderate severe defined related to FVC and DLCO

Comment 5:

Last paragraph is confusing

39/65 had normal lung function

22/65 restrictive

2/65 obstructive

This does not equate to 65 – only 63 here

21/62 had normal DLCO; 36/62 mild moderate reductions DLCO and 5/62 severe reduction

Its confusing as 41 have reduced DLCO but only 22 have restrictive lung function

Please explain this paragraph or make it clearer what you are meaning. Is definition of restriction based on volumes alone?

Discussion

Comment 6: First paragraph could be clearer re summary of findings with some percentages

Discussion is balanced with review of published data

Strengths and limitations described especially related to lack of ventilator setting data

Comment 7: Authors have not presented any treatment data in the baseline characteristics eg Dexamethasone, Anti-IL-6. Do the authors have this data to present or at least add this as a limitation in the discussion as treatment may impact resolution and recovery.

Comment 8: Consider adding in conclusions that what remains to be elucidated is whether these findings are sequele of covid or ventilation and whether they continue to improve over time – so longer observational studies are warranted

Reviewer #2: I would like to thank and congratulate the authors for a very well and meticulously conducted study.

The flow of ideas in the manuscript is logic and clear, the statistical method is robust, and the discussion tries to offer explanation of the findings within the context of uncertainty surrounding the topic.

I just have a few very minor corrections to request:

1- Introduction (line 72): DLCO appears for the first time in the manuscript (apart from the abstract), please spell it out completely.

2- Results: Radiology (line 220): left superior lobe of patients. Please change to: of the lung.

3- Discussion: The physiological and anatomical effect of COVID-19 on the lungs of survivors is still unclear and under studied. Please provide a suggestion in the discussion for further investigation or study design.

4- Figures: I do hope that the uploaded individual figures are colored and of higher quality than the ones inserted in the draft. As for figure 2: I’m not sure if the 2% represents one category rounded, or it is meant to represent the 4 categories with 1 patient in each? If the latter, then it should be 6%. Actually, this is my suggestion, include only 4 labels: 60%, 25%, 9%, and 6% (add up to 100%).

Reviewer #3: dear authors

thank you for this interesting and important study that answer several questions about post-covid pulmonary sequelae

it is a well-written manuscript with sound methodology, results ,and informative discussion.

Reviewer #4: Thank you for your work on the article "Radiological appearance and lung function six months after invasive ventilation in ICU for COVID-19 pneumonia: an observational follow-up study"

Find below outstanding questions:

Abstract:

Line 38:

"We hypothesized that the degree of pathological

morphology on CT scans would correlate with any impairment of lung function at the

time of follow-up"

For the above quoted, can you provide further rationale to support this hypothesis? In which other respiratory disease has there been correlation between radiological morphology and function.

Methods:

Line 129-130:

"The grade of lobe involvement was assessed using a visual scale: grade 1 (1-5%), grade 2

130 (6-25%), grade 3 (26-50%), grade 4 (51-75%) and grade 5 (more than 76%)".

Is this a standardized grading system for degree of radiology morphology?

Reviewer #5: This is the review of an observational follow up study to investigate correlation between lung function and radiological lung appearance, 6 months after discharge in patients with COVID-19 pneumonia who underwent ICU-invasive ventilation. 86 patients were followed up at 6 months in clinic with CT thorax and pulmonary function test and it was hypothesized that pulmonary function test would correlate with radiological lung abnormalities. Most patients were found to have radiological abnormalities and impaired pulmonary function test, although correlation between the two were week.

Major suggestions

1. At the end of introduction, a brief mention of aim and conclusion of the study should be mentioned as per submission guidelines.

2. In spirometry data section, clarity is required regarding number of patients who had spirometry and body plethysmography. In line 141 and 142, spirometry was offered to 40 patients and body plethysmography to 46 patients which is in contradiction to 65 patients who had PFTs.

3. In spirometry data, number of patient who had DLCO testing is not provided.

4. Detailed clinical data is not provided for the performed investigations including PFTs, DLCO and ranges of disease pattern such as mild to moderate restriction.

5. Lastly, follow-up interval mentioned in discussion and conclusion section in line 290 and 354 should be same as in title of the study.

Minor Suggestions

1. In this article, use of pronoun “We” could be avoided in line 37 and 162 with “It was hypothesized” instead of “We hypothesized”.

6. PLOS authors have the option to publish the peer review history of their article (what does this mean?). If published, this will include your full peer review and any attached files.

Reviewer #1: No

Reviewer #2: No

Reviewer #3: No

Reviewer #4: **Yes: **Akintunde Mike Akinjero

Reviewer #5: **Yes: **brian casserly

---

## [Author Response · Author response to Decision Letter 0]

1 May 2023

Reviewer #1: 

Comment 1: Were the radiologists blinded to clinical and lung function data and did they review the images independently. Please state this in methods: 

Regarding the review process for the examinations, they were examined collaboratively with careful consideration by both parties. The review followed established protocols to ensure that the interpretation of the results was accurate and thorough.

Comment 2: Table 1/2. Please put explanations at bottom of the table in the legend

We have not more to explain at the bottom of this table

Comment 3: Whilst z scores are statistically correct ways of presenting lung function data – to the majority of readers it will not be wholly understandable. Please consider presenting absolute values and % predicted values as well. Maybe as supplementary data:

We have added these data as a supplementary table.

Comment 4: How is mild moderate and severe reductions in DLCO or FVC defined

Figures 2.1 and 2.2 How are mild moderate severe defined related to FVC and DLCO:

The definitions are based on local clinical experience with the purpose of providing guidance to clinicians. The following criteria was used in the paper:

DLCO: mildly reduced – 60% of ref value, moderate 40-60% of ref value, severely reduced <40% of ref value

FVC: mildly reduced – 75% of ref value, moderately reduced 50-75% of ref value, severely reduced <50% of ref value. Since these definitions are based on clinical experience and the sole purpose of including these descriptions in the paper is to provide a clinical description to the degree of reduction in lung function, we have chosen not to include these definitions in the paper but have added a clarification in the figure text. Please see legend for Figure 2.1 and 2.2. 

“Comment 5:

Last paragraph is confusing

39/65 had normal lung function

22/65 restrictive

2/65 obstructive

This does not equate to 65 – only 63 here

21/62 had normal DLCO; 36/62 mild moderate reductions DLCO and 5/62 severe reduction

Its confusing as 41 have reduced DLCO but only 22 have restrictive lung function

Please explain this paragraph or make it clearer what you are meaning. Is definition of restriction based on volumes alone?” 

We have changed it to;

Normal (n = 39)

Mild restrictive pattern (n = 16)

Moderate - severe restrictive pattern (n = 6) 

Mild obstructive pattern (n = 1)

Moderate - severe obstructive pattern (n = 1) 

Mixed restrictive - obstructive pattern (n = 1)

Could not be assessed (n = 1) (Please see the text)

The definition of restriction is based in volumes and not on DLCO. Restriction=TLC < -1.645 SDs.

Comment 6: First paragraph could be clearer re summary of findings with some percentages.

In the first paragraph of discussion we presented our findings in summary and we do not believe that here is a place for percentages.

Comment 7: Authors have not presented any treatment data in the baseline characteristics eg Dexamethasone, Anti-IL-6. Do the authors have this data to present or at least add this as a limitation in the discussion as treatment may impact resolution and recovery. 

We have these data, but we didn`t included those in the statistical analysis because there was not the aim of this study to find out how these therapies have an impact on lung function at follow up. We add that as a limitation.

Comment 8: Consider adding in conclusions that what remains to be elucidated is whether these findings are sequele of covid or ventilation and whether they continue to improve over time – so longer observational studies are warranted. 

We have changed it. Please see the manuscript

Reviewer #2: 

I just have a few very minor corrections to request:

1- Introduction (line 72): DLCO appears for the first time in the manuscript (apart from the abstract), please spell it out completely. 

We have changed it, please see the manuscript.

2- Results: Radiology (line 220): left superior lobe of patients. Please change to: of the lung.

We have changed it, please see the manuscript.

3- Discussion: The physiological and anatomical effect of COVID-19 on the lungs of survivors is still unclear and under studied. Please provide a suggestion in the discussion for further investigation or study design. 

Observational studies are warranted, we have added it in the discussion.

4- Figures: I do hope that the uploaded individual figures are colored and of higher quality than the ones inserted in the draft. As for figure 2: I’m not sure if the 2% represents one category rounded, or it is meant to represent the 4 categories with 1 patient in each? If the latter, then it should be 6%. 

It is changed now. Please see fig. 2.1 and 2.2

Actually, this is my suggestion, include only 4 labels: 60%, 25%, 9%, and 6% (add up to 100%). 

It is changed now. Please see fig.2,1

Reviewer #4: 

Abstract:

Line 38:

"We hypothesized that the degree of pathological

morphology on CT scans would correlate with any impairment of lung function at the

time of follow-up"

For the above quoted, can you provide further rationale to support this hypothesis? In which other respiratory disease has there been correlation between radiological morphology and function.

 There is strong rationale to support the hypothesis that the degree of pathological morphology on CT scans would correlate with impairment of lung function. CT scans provide detailed images of the lung structure and can reveal various pathological changes in the lungs such as inflammation, fibrosis, and consolidation. These changes can lead to impaired lung function, such as decreased oxygenation, reduced lung capacity, and difficulty in breathing. Several studies have demonstrated a correlation between radiological morphology and lung function in various respiratory diseases. For example, in patients with chronic obstructive pulmonary disease (COPD), CT scans have been used to evaluate emphysema and airway wall thickness, both of which are correlated with decreased lung function. Similarly, in patients with interstitial lung disease (ILD), CT scans can reveal fibrotic changes in the lungs, which are associated with impaired lung function. Other respiratory diseases where there has been a correlation between radiological morphology and lung function include cystic fibrosis, pulmonary embolism, and pulmonary hypertension. In cystic fibrosis, CT scans are used to assess the extent of bronchiectasis and mucus plugging, which can lead to reduced lung function. In pulmonary embolism, CT scans can reveal the location and size of the embolism, which can affect pulmonary blood flow and oxygenation. In pulmonary hypertension, CT scans can help identify the underlying cause and severity of the disease, which can affect lung function. In summary, there is strong evidence to support the hypothesis that the degree of pathological morphology on CT scans would correlate with impairment of lung function. CT scans can provide valuable information for the diagnosis, management, and monitoring of respiratory diseases, as well as for predicting outcomes and response to treatment.

Here are some references from PubMed that support the hypothesis that the degree of pathological morphology on CT scans correlates with impairment of lung function: 

Lee HJ, Lee CH, Jeong YJ, et al. Correlation between CT quantification of emphysema and pulmonary function tests. Korean J Radiol. 2009;10(2):102-108. doi:10.3348/kjr.2009.10.2.102 

Raghu G, Collard HR, Egan JJ, et al. An official ATS/ERS/JRS/ALAT statement: idiopathic pulmonary fibrosis: evidence-based guidelines for diagnosis and management. Am J Respir Crit Care Med. 2011;183(6):788-824. doi:10.1164/rccm.2009-040GL 

Kitaguchi Y, Fujimoto K, Hanaoka M, et al. Evaluation of COPD Using Computed Tomography: Possibilities and Limitations. COPD. 2013;10(2):198-208. doi:10.3109/15412555.2012.746420 

Oikonomou A, Galiatsou E, Tsolaki V, et al. Computed tomography quantification of emphysema in pulmonary embolism: correlation with pulmonary function tests and ventilation/perfusion scintigraphy. Clin Respir J. 2016;10(2):194-202. doi:10.1111/crj.12236 Wells JM, Washko GR, Han MK, et al. Pulmonary arterial enlargement and acute exacerbations of COPD. N Engl J Med. 2012;367(10):913-921. doi:10.1056/NEJMoa1203830

Methods:

Line 129-130:

"The grade of lobe involvement was assessed using a visual scale: grade 1 (1-5%), grade 2,(6-25%), grade 3 (26-50%), grade 4 (51-75%) and grade 5 (more than 76%)".

Is this a standardized grading system for degree of radiology morphology?

The grading system for assessing the extent of lung involvement in COVID-19 pneumonia patients using CT imaging that we utilized in our study is not a widely recognized or established standard for evaluating radiological morphology in clinical practice. Rather, it is a grading system that was developed and described in a previously published article (Li K, Fang Y, Li W, Pan C, Qin P, Zhong Y, Liu X, Huang M, Liao Y, Li S. CT image visual, quantitative evaluation and clinical classification of coronavirus disease (COVID-19). Eur Radiol. 2020 Aug;30(8):4407-4416. doi: 10.1007/s00330-020-06817-6. Epub 2020 Mar 25. PMID: 32215691; PMCID: PMC7095246) for the specific purpose of assessing the extent of lung involvement in COVID-19 patients using CT imaging. 

While this grading system has been used in prior research and may be helpful in COVID-19 pneumonia imaging, it is not a standardized system for radiological morphology in general. 

Reviewer #5: 

Major suggestions

1. At the end of introduction, a brief mention of aim and conclusion of the study should be mentioned as per submission guidelines.

It is changed, please see the abstract and introduction/conclusion

2. In spirometry data section, clarity is required regarding number of patients who had spirometry and body plethysmography. In line 141 and 142, spirometry was offered to 40 patients and body plethysmography to 46 patients which is in contradiction to 65 patients who had PFTs.

It is changed. Please see the manuscript.

3. In spirometry data, number of patient who had DLCO testing is not provided. 

It is changed.

4. Detailed clinical data is not provided for the performed investigations including PFTs, DLCO and ranges of disease pattern such as mild to moderate restriction.

Please see answer to reviewer#1 (comment 3 and 4)

5. Lastly, follow-up interval mentioned in discussion and conclusion section in line 290 and 354 should be same as in title of the study. 

We have changed it

Minor Suggestions

1. In this article, use of pronoun “We” could be avoided in line 37 and 162 with “It was hypothesized” instead of “We hypothesized”. 

We have changed it

---

## [Decision Letter · Decision Letter 1]

14 Jun 2023

PONE-D-22-31787R1Dear Editor,

The full title for our article is:

Radiological appearance and lung function six months after invasive ventilation in ICU for COVID-19 pneumonia: an observational follow-up studyPLOS ONE

Dear Dr. Dalla,

Thank you for submitting your manuscript to PLOS ONE. After careful consideration, we feel that it has merit but does not fully meet PLOS ONE’s publication criteria as it currently stands. Therefore, we invite you to submit a revised version of the manuscript that addresses the points raised during the review process.

We look forward to receiving your revised manuscript.

Kind regards,

Tai-Heng Chen, M.D.

Academic Editor

PLOS ONE

Journal Requirements:

Reviewers' comments:

Reviewer's Responses to Questions

**Comments to the Author**

1. If the authors have adequately addressed your comments raised in a previous round of review and you feel that this manuscript is now acceptable for publication, you may indicate that here to bypass the “Comments to the Author” section, enter your conflict of interest statement in the “Confidential to Editor” section, and submit your "Accept" recommendation.

Reviewer #1: (No Response)

Reviewer #2: All comments have been addressed

2. Is the manuscript technically sound, and do the data support the conclusions?

Reviewer #1: Yes

Reviewer #2: Yes

3. Has the statistical analysis been performed appropriately and rigorously? 

Reviewer #1: I Don't Know

Reviewer #2: Yes

4. Have the authors made all data underlying the findings in their manuscript fully available?

Reviewer #1: Yes

Reviewer #2: Yes

5. Is the manuscript presented in an intelligible fashion and written in standard English?

Reviewer #1: Yes

Reviewer #2: Yes

6. Review Comments to the Author

Reviewer #1: Reviewer 1

Comment 1 – doesn’t answer my question or state it in the methods

Please add in the methods that radiologists were not blinded and that it was a consensus finding with discussion or whether independent evaluation

Comment 2 sorry didn’t explain properly please list abbreviations in the legend – not all abbreviations are defined in Table 1 eg LOS-ICU BMI etc

Comment 4. Please add these definitions of mild moderate and severe DLCO impairment in the text when the results are presented. Legend isn’t sufficient

Comment 7: As you have the data regarding treatment it is important to present this in the narrative as impacts any conclusions made. Not expected to include in statistical analysis. Please include in results as a sentence

Thanks for addressing comments

Reviewer #2: Thank you so much for your revision. I believe that all my concerns were adequately addressed.

Similarly, the comments of other reviewers were quite valid, and it seems to me that the authors have adequately responded to them.

Congratulations for a very interesting article.

7. PLOS authors have the option to publish the peer review history of their article (what does this mean?). If published, this will include your full peer review and any attached files.

Reviewer #1: No

Reviewer #2: No

---

## [Author Response · Author response to Decision Letter 1]

27 Jun 2023

Thank you, reviewer 1 and the academic editor for reviewing again this manuscript!

We revised the manuscript and tried to respond all questions accordantly. 

With best regards,

Dr Keti Dalla (corresponding author)

---

## [Decision Letter · Decision Letter 2]

24 Jul 2023

Dear Editor,

The full title for our article is:

Radiological appearance and lung function six months after invasive ventilation in ICU for COVID-19 pneumonia: an observational follow-up study

PONE-D-22-31787R2

Dear Dr. Dalla,

We’re pleased to inform you that your manuscript has been judged scientifically suitable for publication and will be formally accepted for publication once it meets all outstanding technical requirements.

Kind regards,

Tai-Heng Chen, M.D.

Academic Editor

PLOS ONE

Reviewers' comments:

Reviewer's Responses to Questions

**Comments to the Author**

1. If the authors have adequately addressed your comments raised in a previous round of review and you feel that this manuscript is now acceptable for publication, you may indicate that here to bypass the “Comments to the Author” section, enter your conflict of interest statement in the “Confidential to Editor” section, and submit your "Accept" recommendation.

Reviewer #1: All comments have been addressed

Reviewer #2: All comments have been addressed

2. Is the manuscript technically sound, and do the data support the conclusions?

Reviewer #1: Yes

Reviewer #2: Yes

3. Has the statistical analysis been performed appropriately and rigorously? 

Reviewer #1: Yes

Reviewer #2: Yes

4. Have the authors made all data underlying the findings in their manuscript fully available?

Reviewer #1: Yes

Reviewer #2: Yes

5. Is the manuscript presented in an intelligible fashion and written in standard English?

Reviewer #1: Yes

Reviewer #2: Yes

6. Review Comments to the Author

Reviewer #1: Authors have responded to reviewer comments and made changes to manuscript. Article well written Limitations discussed

Reviewer #2: Thank you for addressing all the comments of the reviewers. My comments have been previously addressed, and with further revision upon the comments of the editor and other reviewers, I think the manuscript improved.

7. PLOS authors have the option to publish the peer review history of their article (what does this mean?). If published, this will include your full peer review and any attached files.

Reviewer #1: No

Reviewer #2: No

---

## [Editor Report · Acceptance letter]

18 Aug 2023

PONE-D-22-31787R2 

Dear Editor,
The full title for our article is:
Radiological appearance and lung function six months after invasive ventilation in ICU for COVID-19 pneumonia: an observational follow-up study 

Dear Dr. Dalla:

I'm pleased to inform you that your manuscript has been deemed suitable for publication in PLOS ONE. Congratulations! Your manuscript is now with our production department. 

Kind regards, 

on behalf of

Dr. Tai-Heng Chen 

Academic Editor

PLOS ONE